# Proteomic Profiling of Fallopian Tube-Derived Extracellular Vesicles Using a Microfluidic Tissue-on-Chip System

**DOI:** 10.3390/bioengineering10040423

**Published:** 2023-03-27

**Authors:** Didi Zha, Sagar Rayamajhi, Jared Sipes, Angela Russo, Harsh B. Pathak, Kailiang Li, Mihaela E. Sardiu, Leonidas E. Bantis, Amrita Mitra, Rajni V. Puri, Camille V. Trinidad, Brian P. Cain, Brett C. Isenberg, Jonathan Coppeta, Shannon MacLaughlan, Andrew K. Godwin, Joanna E. Burdette

**Affiliations:** 1Department of Pharmaceutical Sciences, College of Pharmacy, University of Illinois at Chicago, Chicago, IL 60607, USA; 2Department of Pathology and Laboratory Medicine, University of Kansas Medical Center, Kansas City, KS 66160, USA; 3Department of Biostatistics and Data Science, University of Kansas Medical Center, Kansas City, KS 66160, USA; 4Kansas Institute for Precision Medicine, University of Kansas Medical Center, Kansas City, KS 66160, USA; 5Charles Stark Draper Laboratory, Cambridge, MA 02139, USA; 6Department of Obstetrics and Gynecology, College of Medicine, University of Illinois at Chicago, Chicago, IL 60607, USA

**Keywords:** fallopian tube, extracellular vesicles, proteomics, microfluidic culture, digital spatial imaging

## Abstract

The human fallopian tube epithelium (hFTE) is the site of fertilization, early embryo development, and the origin of most high-grade serous ovarian cancers (HGSOCs). Little is known about the content and functions of hFTE-derived small extracellular vesicles (sEVs) due to the limitations of biomaterials and proper culture methods. We have established a microfluidic platform to culture hFTE for EV collection with adequate yield for mass spectrometry-based proteomic profiling, and reported 295 common hFTE sEV proteins for the first time. These proteins are associated with exocytosis, neutrophil degranulation, and wound healing, and some are crucial for fertilization processes. In addition, by correlating sEV protein profiles with hFTE tissue transcripts characterized using GeoMx^®^ Cancer Transcriptome Atlas, spatial transcriptomics analysis revealed cell-type-specific transcripts of hFTE that encode sEVs proteins, among which, FLNA, TUBB, JUP, and FLNC were differentially expressed in secretory cells, the precursor cells for HGSOC. Our study provides insights into the establishment of the baseline proteomic profile of sEVs derived from hFTE tissue, and its correlation with hFTE lineage-specific transcripts, which can be used to evaluate whether the fallopian tube shifts its sEV cargo during ovarian cancer carcinogenesis and the role of sEV proteins in fallopian tube reproductive functions.

## 1. Introduction

The human fallopian tube (hFT) is not merely a conduit for the mechanical transport of oocytes from the site of ovulation to the site of implantation, but also a versatile organ that secretes tubal fluid to maintain a physiologically optimized environment for sperm capacitation, fertilization, and early embryo development [1]. The establishment of a successful pregnancy requires appropriate signal exchange between the oviduct and gametes/early embryos, to provide protection from environmental stress and immune response. The dysregulation of this communication is associated with high rates of early pregnancy loss [2,3,4,5].

The epithelium of the human fallopian tube (hFTE) is the source of most high-grade serous ovarian cancers (HGSOCs), the most common and fatal type of epithelial ovarian cancer. hFTE is composed of two major morphologically distinct epithelial cell types: ciliated cells and secretory cells. The secretory cells produce nutrient-rich oviductal fluid and are commonly believed to be the cells of origin for the majority of ovarian carcinomas initiated in the fallopian tube [6,7]. Precursor lesions of HGSOCs, such as secretory cell outgrowths (SCOUTs) defined by the loss of PAX2, p53 mutations termed “p53 signatures”, and serous tubal intraepithelial carcinoma (STIC), are present in hFTE and thought to be progenitors of HGSOC [6,8,9]. The tubal paradigm proposes that the STIC present in the fimbriated end of the fallopian tube can become invasive and spread onto the peritoneal surface and the ovary, giving rise to metastatic HGSOC [10]. Evolutionary models predict a 7-year time window between the development of STIC and the initiation of ovarian cancer [6]. This period of transformation offers a significant time window for preventive interventions, including the surgical removal of fallopian tubes. Ongoing clinical trials have shown that salpingectomy, which is the removal of fallopian tubes without the removal of ovaries, is associated with reduced ovarian cancer risk [11]. However, due to a lack of effective biomarkers, current screening strategies have failed to detect ovarian cancer at early stages when survival rates are much higher. As such, there is an urgent and unmet need to expand the class of liquid-based biopsies for ovarian cancer screening to detect ovarian cancer at an early stage, when the treatments are more effective.

Recently, extracellular vesicles (EVs) have been identified in a wide variety of biofluids, including reproductive tract fluids, plasma, urine, and saliva. EVs are nanosized biovesicles surrounded by lipid bilayers, released by almost all living cells. There are different subtypes of EVs, which are broadly categorized based on size and biosynthetic pathways. Small extracellular vesicles (sEVs) are a subset of EVs with a size range of 50–200 nm. These are the representative EV subtypes most widely used in studies for their role in different pathological conditions [12]. Extracellular vesicles represent a general mechanism by which cells modulate intercellular crosstalk to alter the phenotype and characteristics of the neighboring cells [13]. The molecular cargos of EVs, such as miRNAs, proteins, and lipids, can be incorporated into recipient cells to regulate gene expression and signaling. In addition, EVs reflect the molecular content of the parental cells and can be used to monitor disease states. As such, the role of EVs in the development and progression of various types of cancer has been reported. EVs act as a messenger between tumor cells and the host microenvironment, thereby influencing multiple cellular processes, including angiogenesis, coagulation, vascular leakiness, and the reprogramming of host stromal cells to form the pre-metastatic niche (PMN) leading to metastasis [14]. Our group has previously reported the role of EVs in phenotypic conversion of cells to tumor-promoting cells, mediating drug resistance behavior, rescuing stem cells, and PD-L1-based immunotherapy [15,16,17,18,19]. Others have found that the EV proteomic profile can predict the metastatic tropism of certain cancers [20,21]. Therefore, EVs have attracted broad research interest for their clinical potential as novel therapeutic targets and an emerging class of biomarkers.

Within the last decade, most studies on the reproductive function of human fallopian tube EVs have been conducted using oviductal EVs from rodents, domestic, and farm animals, due to the technical limitations associated with developing human experimental models. Oviductal EVs are an essential component of oviductal fluid, supporting gamete function and early embryo development [22]. In the canine reproductive tract, oviductal EVs shuffle miRNAs, such as miR-30b, miR-375, and miR-503, to facilitate follicular growth and oocyte maturation, potentially through WNT, MAPK, and TGFβ signaling [23]. Studies in the porcine oviduct showed that the uptake of oviductal EVs by oocytes can decrease the zona pellucida binding, thereby reducing polyspermy [24]. PMCA4, a plasma membrane calcium pump essential for male fertility, is secreted through murine oviductal EVs and trafficked to sperm to promote hyperactivation and acrosomal reaction [25]. The presence of PMCA4 in EVs from fallopian tube luminal fluid, as reported by Bathala and colleagues, suggests a likely conserved mechanism of PMCA4-mediated acrosomal reaction in humans [26]. Proteomic analysis identified bovine oviductal EV proteins associated with fertilization (ADAM9, GSN, and OVGP1), gamete generation (GNAI2 and STX3), sperm fertilization ability (HSP70), and embryo development (YBX1) [3]. The oviductal EVs collected from the bovine oviduct fluid can increase blastocyst yield and enhance embryo survival and quality in vitro [3]. However, further studies are needed to elucidate the human EV protein cargos and their functions in reproductive biology.

Despite advancements in the scientific understanding of oviductal EVs using farm and domestic animal models, it remains challenging to study the molecular content of fallopian-tube-derived EVs and their roles in human reproductive biology and ovarian cancer initiation. Our study aims were to (1) develop a microfluidic culture method with dynamic media flow conditions and mechanical stimulation, that allows for long-term culture of hFTE tissue to yield enough EVs for proteomic characterization, (2) elucidate the protein cargos of EVs secreted by hFTE tissue explant ex vivo for the first time, (3) compare the expression with other species, (4) describe the baseline expression of cancer-related sEV cargos in normal conditions, and (5) correlate sEV proteomic cargo with ciliated and secretory cell mRNA expression by digital spatial transcriptomics. We cultured fallopian tube primary tissue on the PREDICT-multi-organ-system (PREDICT-MOS) [27] for six days, and the conditioned media from the hFTE culture were collected daily. EVs secreted by hFTE were isolated through ultracentrifugation from conditioned media, and EV protein cargos were purified for proteomics analysis. Additionally, we profiled the transcriptome of secretory and ciliated cells from hFTE cultured in the PREDICT-MOS over 24 h, using the GeoMx^®^ Digital Spatial Profiler. Our study reveals the cell-type-specific transcripts that encode for proteins which are sorted and enriched in hFTE sEVs, and lays a foundation for establishing a baseline proteomics profile of hFTE-derived sEVs, which can be used to evaluate whether the fallopian tube shifts its sEV cargo during ovarian cancer carcinogenesis.

## 2. Materials and Methods

### 2.1. Culturing of Human Fallopian Tube (hFTE) Tissue Explants on the PREDICT-Multi-Organ-System

The studies have been approved by the appropriate institutional research ethics committee and have been performed in accordance with the ethical standards, as laid down in the 1964 Declaration of Helsinki and its later amendments or comparable ethical standards. The use of human samples was approved by the Institutional Review Board under the Human Tissue Bank Protocol at UIC (IRB #2017-0574), or the Biospecimen Repository Core Facility (HSC #5929) at the KUMC. De-identified fallopian tube tissues were obtained from women who provided informed consent.

The tissue was stored in dissection media (MEM/Nutrient Mixture F-12 medium, 1:1) with 10% FBS, and was dissected within 24 h of receival. The epithelium of the fallopian tube was isolated from the underlying stroma and divided into 2 mm × 2 mm sections, as previously described [28]. Following the dissection, fallopian tube explants were cultured overnight in 0.4 µm pore Millicell inserts with growth media (MEM with 0.3% bovine serum albumin, 1% penicillin/streptomycin, and ITS medium). Cilia beating was confirmed under 40X magnification the following day, before the explants were transferred to the PREDICT-MOS tissue plate for dynamic culture. The PREDICT-MOS microfluidic device sustained a continuous media flow of 40 µL/h. The conditioned media were collected, and the fresh media were replenished every 24 h. After the six-day culture, the PREDICT-MOS tissue plate was washed with distilled water and then 5% EtOH for 30 min. The pump unit was surged with 1% tergazyme solution, 70% EtOH, and distilled water, each for 1 h.

### 2.2. Live/Dead Staining

Tissues were transferred from the PREDICT-MOS tissue plate to a 96-well plate after a six-day dynamic culture. Then, 100 µL of 2 × calcein AM/BOBO-3 iodide solution and 100 µL of PBS were added directly to the tissue. After incubating for 15 min at 20–25 °C, the tissues were transferred onto a microscopic slide with coverslips, for imaging under the fluorescent microscope.

### 2.3. Spatial Transcriptomic Analysis to Characterize Epithelial Cell Types of Fallopian Tube Explant Cultured in the PREDICT-Multi-Organ-System

To obtain the transcriptomics profile of fallopian tube secretory and ciliated cell populations cultured in the microfluidic system, hFTE explants were cultured in the PREDICT-MOS microfluidic device for 24 h. The tissues were then fixed in 10% paraformaldehyde for 24 h and transferred to ethanol gradients, before paraffin-embedding and sectioning. For GeoMx^®^ Digital Spatial Profiling (Nanostring, Seattle, WA, USA), the protocol from MAN-10130 for the software v2.1 (RNA Manual Slide Prep Reagents) was followed. In summary, we first used a library of RNA probes (~1800 genes) with a photocleavable linker (oligo tags) to perform in situ hybridization on fallopian tube tissue slides. Next, we used fluorescent antibodies to label ciliated cells and secretory cells. Based on the fluorescent labels, we selected regions of interest (ROI) containing ciliated or secretory cells for analysis. This allows us to collect the transcriptomics data from these two cell populations separately. The slides were exposed to UV light, and the oligo tags were collected for next-generation sequencing. The reads of these oligo tags on next-generation sequencer reflect the abundance of their corresponding RNA probes and original RNA content present in the fallopian tube tissue. The detailed procedures are described as follows.

Briefly, the slides were deparaffinized in xylene, followed by a series of EtOH gradients. Antigen target retrieval was performed in 1x Tris EDTA in a steamer at ~99 °C for 20 min. To expose RNA targets, slides were placed in a staining jar containing 1 µg/mL proteinase K in DEPC water, and incubated in a bead bath at 37 °C for 20 min. To preserve the tissue, tissue was washed in 10% NBF for 5 min, followed by 2 × 5 min washes in an NBF stop buffer. RNA probe hybridization solution was prepared using the GeoMx^®^ Cancer Transcriptome Atlas RNA probe mix (NanoString), following the recommended dilutions. The hybridization solution was added, and the slide was covered with a HybriSlip coverslip and incubated overnight (16–24 h) in a humidified hybridization chamber at 37 °C. For the full procedure, see MAN-10130 for the software v2.1.

The following day, the slides were removed from the hybridization chamber and washed to remove unconjugated probes (2 washes in 1:1 4 × SSC and 100% formamide at 37 °C, followed by two washes in 2X SSX at room temperature). Slides were incubated with PAX8 antibody (Proteintech, conjugated with Alexa Flour 594) and FOXJ1 antibody (Invitrogen, 14-9965-82) for 1 h, followed by 30 min incubation with a secondary antibody against the FOXJ1 conjugated with CY3, and DNA staining with Syto13. On the same day, prepared slides were scanned on the GeoMx^®^ Digital Spatial Profiler. Regions of interest were selected based on the presence of secretory and ciliated cells, and probes for these regions were collected in a 96-well plate (details of ROI selection and collection found in MAN-10116-05 for the software v2.3, Nanostring, Seattle, WA, USA). Probes were sequenced on the Illumina Next Generation Sequencer after library preparation, following the protocol provided by Nanostring in the MAN-10117-05 NGS Readout Library Prep User Manual.

### 2.4. sEV Isolation and Characterization

Conditioned media from hFTE tissue explant cultures were collected, then spun down at 500× *g* for 10 min to remove cells and cell debris, followed by a 2000× *g* spin for 20 min to remove apoptotic bodies. Conditioned media were frozen at −20 °C until about 30–40 mL total was collected, before proceeding to differential ultracentrifugation. Conditioned media were spun at 10,000× *g* for 75 min at 4 °C to remove large EVs. The pellet was discarded, and the supernatant spun at 100,000× *g* for 1.5 h at 4 °C to pellet sEVs. To wash the pellet, most of the supernatant was discarded, and the remainder (~1 mL) was used to resuspend the pellet. This suspension was transferred to smaller ultracentrifuge tubes using a sterile syringe, and the remaining volume was filled with filtered PBS. Following another spin at 100,000× *g* for 1.5 h at 4 °C, the supernatant was removed via aspiration, and the resulting pellet was resuspended in 100–120 µL filtered PBS and frozen at −80 °C until use. Total particle count and average particle size were estimated using the NanoSight LM10 system and accompanying NTA software v2.3 (NanoSight Ltd., Salisbury, UK). All NTA readings were performed at 20 °C, camera level at 12, and detection threshold at 4. The total protein content of sEVs was quantified using the Bradford assay.

### 2.5. ExoView Analysis of sEVs

Purified sEV samples were quantified using ExoView™ Tetraspanin kits (NanoView Bioscience, USA), following the manufacturer’s instructions. sEV samples were diluted (1:1) with incubation solution and loaded on the Nanoview chips for incubation overnight. Each chip included antibodies targeting CD9, CD63, and CD81 EV tetraspanin markers. After binding, the chips were washed three times with the incubation solution, followed by staining with detection antibodies conjugated to fluorescent markers (anti-human CD81-CF^®^ 555, CD63-CF^®^ 647, and CD9-CF^®^ 488A) at manufacturer-provided concentrations. After three more washes, the plate was scanned on the ExoView™ R100 imager and analyzed using the ExoScan 2.5.5 software (NanoView Bioscience, Boston, MA, USA).

### 2.6. Proteomics Analysis of sEV Cargos Secreted by Fallopian Tube Epithelium

Proteomics profiles of sEVs were characterized following a label-free mass spectrometry technique with the data-dependent acquisition method (DDA), at the University of Arkansas for Medical Science Proteomics Core Facility. Based on protein abundance, patients variables were characterized using the Jaccard index and PCA analysis. Pathway analysis was performed using gene ontology (GO)-term biological processes.

sEVs were lysed to release proteins and other cargo using an SDS lysis buffer. sEV proteins (15 µg per sample) were reduced, alkylated, and digested using filter-aided sample preparation [29] with sequencing grade-modified porcine trypsin (Promega, Madison, WI, USA). Tryptic peptides were then separated by reverse-phase XSelect CSH C18 2.5 um resin (Waters, Milford, MA, USA) on an in-line 150 × 0.075 mm column, using an UltiMate 3000 RSLCnano system (Thermo Fisher Scientific, Waltham, MA, USA). Peptides were eluted using a 90 min gradient from 98:2 to 65:35 buffer A:B ratio. Mobile phase A consisted of 0.1% formic acid in 0.5% acetonitrile, and mobile phase B consisted of 0.1% formic acid in 99.9% acetonitrile. Eluted peptides were ionized by electrospray (2.4 kV), followed by mass spectrometric analysis on an Orbitrap Eclipse Tribrid mass spectrometer (Thermo Fisher Scientific). MS data were acquired using the FTMS analyzer in profile mode at a resolution of 120,000 over 375, to 1200 *m*/*z*. Following HCD activation, MS/MS data were acquired using the ion trap analyzer in centroid mode and a normal mass range, with a normalized collision energy of 30%. Proteins were identified by a database search using MaxQuant Version 2.0.3.1 (Max Planck Institute of Biochemistry, Martinsried, Germany), with a parent ion tolerance of 2.5 ppm and a fragment ion tolerance of 0.5 Da. Scaffold Q+S Version 5.1.2 (Proteome Software, Portland, Oregon, USA) was used to verify MS/MS-based peptide and protein identifications. Protein identifications were accepted if they could be established with less than 1.0% false discovery and contained at least two identified peptides. Protein probabilities were assigned by the Protein Prophet algorithm [30].

### 2.7. Immunohistochemistry (IHC) Staining

After the 6-day culture of the fallopian tube on the PREDICT-MOS microfluidic device, the tissue was fixed in 10% paraformaldehyde. Paraffin-embedded slides were deparaffinized by two xylene washes, each for 10 min, followed by rehydration in an ethanol gradient of 100%, 95%, 70%, 50%, and pure ddH_2_O, for 3 min each. Antigen retrieval was performed in 0.1 M Na citrate (pH 6) in a microwave set on high for 2 min, followed by 13 min on low (power 10), after which the slides were permitted to cool for 30 min. Slides were blocked with avidin and biotin, and then with 10% goat serum in 3% BSA-TBS solution for 1 h. Primary antibodies to PAX8 (Proteintech, #10336-1-AP), FOXJ1 (Invitrogen, #14-9965-82), VCAN (Invitrogen, S351-23), CD44 (Abcam, ab119348), HSP90 (Santa Cruz Biotech, sc-13119), and FLNA (Proteintech, 67133-1-Ig) were incubated overnight at 4 °C. The slides were washed in 0.1% TBS-T thrice, each for 5 min, and incubated in secondary antibodies conjugated to biotin in 3% BSA-TBS for 30 min the following day. After repeating the 0.1% TBS-T wash thrice, the slides were incubated in the ABC reagent for 30 min. The DAB substrate was added to the slides in the presence of HRP and developed for 3–10 min. The reaction was quenched by ddH_2_O, and the tissues were counterstained with hematoxylin.

### 2.8. Transmission Electron Microscopy (TEM)

Purified sEVs were diluted in filtered PBS for use in TEM. Freshly thawed sEV samples were used for imaging. Glow-discharge-treated and carbon-film-coated 300-mesh copper grids were floated in 20 µL of purified sEV solution for 20 min. The grids were washed in six successive water droplets and stained in a droplet of 1% uranyl acetate for 5 s. Grids were dried for 15 min and imaged on a JEOL JEM-1400 TEM. All imaging was completed within two days of the initial stain. The JEOL JEM-1400 transmission electron microscope was purchased with funds from the NIH grant 1S10RR027564.

### 2.9. Statistics

Statistical analysis was performed on Microsoft^®^ Excel^®^ for Microsoft 365 MSO (Version 2202, Microsoft, Redmond, WA, USA) and GraphPad Prism. Data are presented as the mean +/− s.e.m. For heatmap analysis, rows were centered, and unit variance scaling was applied to the rows. Both the rows and columns were clustered by Pearson’s correlation as a method, and Ward as a distance. The volcano plot was created using the GeoMx DSP Analysis Suite (Version 2.4.0.421) by using a two-sided, unpaired *t*-test to compare transcript expression in the ciliated and secretory regions of interest (ROIs).

### 2.10. Reagents and Kits Used

The following reagents and kits were used in this experiment: live/dead staining (#R37601, Invitrogen, Waltham, MA, USA), RPMI 1640 medium 1× (Cat. N.: SH30027.01, Cytiva, Tokyo, Japan), DPBS (Dulbecco’s phosphate buffered saline, ref. no. 21-031-CV, Corning Inc., Corning, NY, USA), DMEM/F-12 50/50, 1× (Dulbecco’s mod. eagle’s medium/Ham’s F-12 50/50 Mix, without L-glutamine, Corning, ref. no. 15-090-CV), FBS (fetal bovine serum, Cat. No.: S11150, Atlanta Biologicals^®^, Flowery Branch, GA, USA), pen/strep (penicillin–streptomycin, 10,000 units/mL penicillin, 10,000 µg/mL streptomycin, gibco^®^, ref. no. 15140-122), insulin (recombinant human insulin, gibco^®^, formula no. A11382IJ), UCG (Ultroser™ G, serum substitute for animal cell culture, PALL Life Sciences, 20 mL, ref. no. 15950-017), Exoview Kit, Avidin/Biotin Blocking Kit (SP-2001, Vector Labs, Burlingame, CA, USA), VECTASTAIN^®^ ABC-HRP Kit, peroxidase (Vector Labs, PK-4000), DAB Substrate Kit, peroxidase (HRP), and nickel, (3,3′-diaminobenzidine) (Vector Labs, SK-4100).

## 3. Results

### 3.1. Primary Fallopian Tube Explant Maintains Epithelial Architecture and Distinct Cell Subtypes during Long-Term Culture in the Dynamic Organ-On-Chip System

The organ-on-chip microphysiological system can maintain the in-vivo-like morphology and function of the oviduct epithelial cells compared to the 2D static culture, where cells tend to rapidly lose polarity, compromising their cilia and secretory properties [31]. We initially developed a microfluidic organ-on-chip system to support an ex vivo culture of human reproductive tract organs, and have now further engineered the system for the culture of fallopian tube tissue and ovarian explants, as described by Russo and colleagues [27]. To optimize the PREDICT-MOS microfluidic device for the collection of sEVs, human fallopian tube epithelium (hFTE) explants were cultured in the PREDICT-MOS for six days, and conditioned media containing fallopian tube sEVs were collected every 24 h. (Figure 1A). The viability of the tissue was assessed through live/dead staining at the end of the six-day culture. Based on the cell membrane integrity, live cells emitted a green fluorescence and dead cells with a damaged cell membrane produced a nuclear red fluorescence. The abundant presence of green signal and a lack of red signal in the tissue shows that the majority of the fallopian tube epithelial cells in the hFT explants are viable after the six-day culture on the microfluidic device. The tissue was then fixed in 10% formalin and paraffin-embedded for immuno-histological analysis. IHC staining for FOXJ1 and PAX8, the respective markers for ciliated and secretory cells, showed the presence of two major morphologically distinct epithelial cell populations after culture (Figure 1B).

### 3.2. Physical and Molecular Characterization of Small Extracellular Vesicles (sEVs) Derived from hFTE Tissue Explants Cultured in the PREDICT-MOS

sEVs were isolated from conditioned media collected from hFTE tissue explants cultured in the PREDICT-MOS for six days, using differential ultracentrifugation. The mean size of sEVs from seven different patient tissue ranged from 153 to 216 nm, while the mode size ranged from 102 to 157 nm, as quantified by nanoparticle-tracking analysis (NTA) (Figure 2A). The total protein content showed that the EV protein amount ranged from 16 µg to 104 µg and the total EV number varied from 7.3 × 10^9^ to 4.0 × 10^10^ between different patient samples (Figure 2B). Morphological analysis of sEVs was carried out using electron microscopy, which showed a heterogenous population of spherical and cup-shaped vesicular structures (Figure 2C). Molecular characterization was carried out using single-EV analysis Exoview platform to detect the presence of EV marker proteins: CD9, CD81, and CD63. Exoview analysis showed the presence of all three tetraspanins (CD9, CD63, and CD81). Representative images of sEVs captured with the CD63 probe are reported in Figure 2D, with the detection of CD9 (blue), CD63 (red), and CD81 (green) proteins, as represented by the blue, red, and green puncta. Quantitative analysis showed the presence of all three tetraspanins in each of CD9, CD63, and CD81 captured sEV, suggesting the colocalization of tetraspanin markers in single sEVs (Figure 2E). Mouse IgG was used as a negative control, which showed minimal signal compared to sEVs. Physical and molecular characterization demonstrated the size, total protein, total number, and morphological structure of sEVs derived from the PREDICT-MOS platform, thereby validating the successful isolation of sEVs with characteristics similar to sEVs derived from the conventional 2D cell culture. The significant variation in total protein content and sEV number among different patient samples may represent patient-specific variations in sEV production.

### 3.3. Proteomics Profile of hFTE sEVs

With the physical and molecular characterization of sEVs, we mapped the proteomics profile of sEVs secreted by hFTE tissue explants in the PREDICT-MOS, under dynamic media flow conditions. sEVs (*n* = 7) were digested in solution, following filter-aided sample preparation resulting in tryptic peptides. Peptides were analyzed using tandem mass spectrometry combined with liquid chromatography (LC-MS/MS), and proteins were identified by a database search using MaxQuant (Max Planck Institute). In total, 254 (lowest) to 2010 (highest) proteins were identified, which varied primarily based on individual tissue sources (Appendix A). Contaminated proteins (proteins of non-human origin), keratins, albumins, and proteins identified as clusters of proteins (aggregates) were filtered out from the total proteins. Next, we selected 295 proteins that were identified in at least five out of seven biological replicates for subsequent analysis.

These 295 proteins present in hFTE sEVs were cross-referenced with two publicly available extracellular proteome databases: ‘Vesiclepedia’ and ‘Exocarta’. This comparison showed a considerable overlap of 93%, suggesting the majority of identified hFTE sEV proteins from the PREDICT-MOS platform shared commonality with pre-identified EV proteins (Figure 3A). Nineteen exclusively identified proteins in our study were not documented to be transported by EVs (source species: *Homo sapiens*) previously, based on these two databases (Appendix A). Fifteen of the 19 proteins were nuclear (histone) proteins, and only five proteins were non-nuclear proteins, including ATP synthase subunit beta (mitochondrial, ATP5F1B), maltase-glucoamylase 2 (MGAM2), receptor of activated protein C kinase 1 (RACK1), skin-specific protein 32 (XP32), and ATP synthase subunit alpha (ATP5F1A). These proteins are involved in biological processes such as mitochondrial ATP synthesis coupled proton transport, co-translational protein targeting to membrane, angiostatin binding, and MHC class I protein binding. The presence of H2AX nuclear protein, one of the unique sEV proteins identified in our database, may allow for the examination of the extent of nuclear damage in the secreting cells. The phosphorylated form of H2AX can be used as a molecular marker of DNA damage and repair. The phosphorylation of the Ser-139 residue of the histone variant H2AX is an early cellular response to nuclear damage induced by DNA double-strand breaks [32].

The Jaccard plot shows the similarity between identified sEV proteins among seven different biological replicates (Figure 3B). The similarity between biological samples is represented by a coefficient ranging from 0 to 1, which refers to 0% to 100% similarity or overlap between identified proteins. Five biological replicate samples showed high similarity, with a 0.9 similarity coefficient, while two biological replicates showed a low similarity coefficient of 0.4–0.6. This depicts the individual-specific variation in hFTE sEVs among the biological replicates, which was further supported by heatmap analysis of 295 sEV protein abundance among seven biological replicates representing unique patient samples. Figure 3C shows the heatmap with hierarchical clustering depicting the relative protein abundance of 295 hFTE sEV proteins among seven biological replicates. Biological replicate samples 1 and 4 have a high expression/abundance of proteins compared to other samples, as demonstrated by the high intensities (red) of the heatmaps. Next, we analyzed the presence of canonical EV proteins often identified in sEVs and represented as EV marker proteins, including tetraspanins (CD63, CD9, CD81), heat shock proteins, flotillin, annexins, syntenin, and others. Figure 3D shows the heatmap of a relative abundance of EV marker proteins among seven biological replicates. The consistent presence of EV marker proteins among seven different biological replicates provides additional evidence of the presence of EVs following our microfluidic-based sEV isolation method. Gene ontology (GO) analysis showed that 295 hFTE sEV proteins are associated with EV-related biological processes such as exocytosis, and REACTOME pathway analysis showed association with pathways related to vesicle-mediated transport, membrane trafficking, and post-translational protein modification (Figure 2E,F). In addition, EV proteins were found to be involved with biological processes related to immune response, including leukocyte-mediated immunity, neutrophil degranulation, and wound healing.

### 3.4. The Comparison of Fallopian Tube sEV Protein Content across Different Species and with Benign Fallopian Tube and STIC Lesions in the Tissue

Literature analysis shows a list of sEV proteins from hFTE critical for FT reproductive functions and ovarian cancer progression, which have been summarized in Table 1 and Table 2, respectively. Oviductal EV protein cargos have been previously characterized in porcine, bovine, and feline oviductal fluids [3,33,34]. The 295 sEV proteins identified from hFTE cultured in dynamic conditions were compared with oviduct EV proteins detected in farm and domestic animals (Figure 4A). sEV protein cargos detected from hFTE and bovine oviduct had the highest similarity, with a Jaccard index of 0.198 (Appendix A). Thirty-four proteins are present in all species, including annexin family proteins (ANXA1, ANXA2, and ANXA5) and heat shock family proteins (HSPA8, HSP90) associated with exocytosis and cellular secretion processes. Of the proteins, 119 were only detected in hFTE-secreted sEVs. The GO biological process revealed that these sEV proteins are related to leukocyte-mediated immunity, neutrophil degranulation, and regulated exocytosis (Appendix A). Since EV cargos can reflect the content of parental cells, sEV proteins secreted by serous tubal intraepithelial carcinoma (STIC) fallopian tube tissue can be potentially indicative of early lesion of ovarian cancer in the fallopian tube. Acland et al. conducted a proteomics profiling of the STIC and the adjacent benign fallopian tube tissue from the same patient, and detected 273 proteins that were exclusively expressed in STIC, but not the benign fallopian tube [35]. We compared hFT sEV protein cargos with proteins that are only expressed in STIC tissue identified in the Acland study, to begin to predict which proteins expressed in primary STIC tissue might be packaged into sEVs. The results indicated 32 fallopian tube sEV proteins detected in our study are also expressed in STIC tissue exclusively (Figure 4B). These proteins include ALDH2, a cancer stem cell marker associated with invasion and metastasis [36]; RAN, a small GTPase that contributes to cancer proliferative signaling, chemoresistance, and metastasis [37]; CLIC1, a chloride pump that is upregulated in cancer to promote tumor invasion, metastasis, and angiogenesis [38]; and UBA1, the dominant E1 enzyme that facilitates DNA repair through the BRCA1- and TP53BP1-associated mechanism (Appendix A) [39].

### 3.5. Correlation of hFTE sEV Proteomics Profile with Cancer-Associated hFTE Tissue Explant Transcriptome

Next, we evaluated the correlation of hFTE sEV proteins with the cancer-associated transcriptome profile of hFTE tissue explants, using GeoMx™ digital spatial profiling. GeoMx™ RNA assays allowed for the profiling of the whole transcriptome or selected probe sets reflecting specific biology, using single formalin-fixed paraffin-embedded tissue sections, with digital spatial profiling. We used the GeoMx^®^ Cancer Transcriptome Atlas (~1800 genes) to profile the transcriptome of hFTE tissue explants. The Cancer Transcriptome Atlas is designed for the comprehensive profiling of tumor biology, the tumor microenvironment, and the immune response with spatial resolution. We identified the presence of 1480 transcripts in hFTE tissue explants cultured in the PREDICT-MOS, with the ability to distinguish ciliated and secretory cell transcripts. We compared this 1480-transcript profile with the 295-sEV proteome profile, and found 61 common gene IDs, suggesting that 61 transcripts were transcribed to proteins and sorted in sEVs (Figure 5A). Protein interaction via STRING shows that these proteins have a strong interaction with each other (PPI enrichment *p*-value: < 1.0 × 10^−16^) and are biologically connected as a group (Figure 5B). Out of 61 proteins, 15 proteins were membrane proteins that can be potentially used as target proteins for sEV capture and isolation. These 15 membrane proteins include CD9, CD44, CD47, CD59, CD63, CD81, NT5E, ITGB1, LAMP2, THY1, SLC2A1, DPP4, HLA-DRA, HLA-A, and HLA-G (Appendix A).

Gene enrichment analysis via FUNRICH reveals major molecular functions, biological processes, Reactome pathways, and cellular components associated with the 61 proteins (Figure 5C). The major molecular functions include enzyme binding, signal receptor binding, RAGE (receptor for advanced glycation end-products) receptor binding, and calcium-dependent protein binding. Cellular component analysis shows that these 61 sEV proteins are associated with varied cellular components, including cytosol, cytoskeletal, plasma membrane, and extracellular exosome. Major biological processes associated with 61 hFTE sEV proteins include the regulation of the cell shape, platelet aggregation, cell-matrix adhesion, and positive regulation of stress fiber assembly. Likewise, Reactome pathway analysis showed that the major pathways associated with 61 hFTE sEV proteins are immune-response-related pathways, including neutrophil degranulation, platelet degranulation, G-protein-coupled estrogen receptor-1 (GPER1) signaling, metal sequestration by antimicrobial proteins, and RHO GTPase-activated NADPH oxidases.

The secretory cells in the FT epithelium are commonly believed to be the cells of origin for the majority of ovarian carcinoma initiated in the fallopian tube. The loss of ciliated tubal cells is considered a risk for ovarian cancer [63]. Among the 61 sEV proteins, we analyzed if they have differential expression profiles in the ciliated and secretory cells of fallopian tube tissue. Figure 5D shows the volcano plot of differential gene expression of secretory vs. ciliated cells in benign fallopian tube tissue explants (six biological replicates), cultured for 24 h in the PREDICT-MOS platform. Four (4) transcripts that are relatively upregulated (*p*-value < 0.05, |log_2_fold change| > 0.5) in secretory cells were present in the hFTE sEV proteomics profile: *FLNA* (filamin-A), *TUBB* (tubulin beta-chain), *FLNC* (filamin-C), and *JUP* (junction plakoglobin), while no transcripts relatively upregulated in ciliated cells were present in the hFTE sEV proteomics profile. FLNA and FLNC are actin crosslinking proteins that play a role in cellular cytoskeletal organization and interact with diverse proteins, including transmembrane proteins, signaling molecules, and DNA-damage repair proteins [64]. TUBB is the major constituent of microtubules and is involved in GTP binding and nucleotide binding. JUP plays a role in the arrangement and function of cytoskeletal proteins within the cell. It is of interest that the protein products of these four genes overexpressed in the secretory cells are packaged into sEVs, as FT secretory cells are commonly considered to be the progenitor cells of high-grade serous ovarian cancer. We validated the expression of some of the proteins from the list of 61 sEV proteins: FLNA, CD44, VCAN, and HSP90 in fallopian tube tissue, using immunohistochemistry (IHC), which showed positive signals for all the markers tested (Figure 5E). Using multiple-approach digital spatial profiling, proteomics, and tissue IHC, we have sorted and identified 61 sEV proteins which may be listed as “EV biomarkers to watch” at different stages of disease progression leading to high-grade serous ovarian cancer.

## 4. Discussion

Extracellular vesicles secreted by oviductal cells are an essential component of oviduct fluid, improving the fertilization process and embryo quality, as shown by several studies conducted in mammals [3,26,65]. The proteomics profiling of oviduct EV cargos has been conducted using bovine, feline, and porcine oviduct fluids [3,33,34]. However, the direct characterization of EV content from human fallopian tubes was not feasible due to the limitation of biomaterials and invasive fallopian tube fluid collection procedures. Therefore, we have optimized an organ-on-chip perfusion system to culture the fallopian tube ex vivo for the collection of sEVs, that yields adequate protein for mass spectrometry analysis. We conducted the first proteomics characterization of human FT-derived sEVs and have reported 295 sEV proteins identified in at least five out of seven patient samples (biological replicates). The top biological processes associated with these proteins are exocytosis, leukocyte-mediated innate immunity, neutrophil degranulation, and wound healing.

Oviduct EVs can be incorporated into the sperm, oocyte, and embryo to assist oviductal reproductive functions [22]. We have identified FT sEV proteins, and many of them have reported roles for supporting a proper microenvironment for fertilization and preimplantation embryo development in the fallopian tube. Human spermatozoa require an incubation period in the fallopian tube to acquire fertilization capacity. We detected FT sEV proteins, such as PMCA4 and GSN, that regulate sperm capacitation and acrosome reaction, which is essential for sperm–oocyte interaction and fertilization. Heat shock proteins, such as HSP70 and HSP90, can additionally potentially improve sperm fertilization capacity and modulate sperm capacitation. TUBB4B secreted by FT through sEVs is an essential component of the cilia that regulates cilia motility for oocyte pickup and facilitates gametes transport [48]. Proteins crucial to cell fate decision and cell division during early embryogenesis, such as ADAM10 and PFN1, were detected in human FT sEVs [42,66]. The protein cargos of FT sEVs also contain immunosuppressive proteins PZP and HLA-G, that protect the fetus against the maternal immune system. Notably, we did not detect OVGP1 (oviduct-specific glycoprotein) in sEVs derived from FT cultured in our microfluidic model, as reported by Almiñana and colleagues [3]. This is probably due to the lack of hormonal stimulation. Specifically, we did not add estradiol in our ex vivo hFTE culture. Of the proteins, 119 were exclusively detected in human fallopian tube-derived sEVs, but none in the other mammalian species analyzed. This finding revealed the unique human fallopian tube sEV biology compared mammals. Among these human-unique sEV proteins, CD47 is a leukocyte surface antigen important for immune cell recruitment to the site of infection [67]. Human leukocyte antigen G (HLA-G) is involved in the protection of the fetus against the maternal immune system by promoting immune tolerance [68]. Lysozyme (LYZ) is a bactericidal enzyme secreted through neutrophil degranulation against foreign pathogens [69]. Myeloperoxidase (MPO) is one of the most abundant proteins in neutrophils [70]. Ras oncogene family proteins RAB5B, RAB6A, and RAP1A, are potentially involved in synaptic vesicle exocytosis [71].

Extracellular vesicles from cancer cells modulate intercellular communication to alter the phenotype of the neighboring cells. Cancer cells commonly secrete more EVs than healthy cells and shuttle cancer-promoting factors to their EVs which helps to promote tumorigenesis [72,73]. Tumor-derived EVs drive the formation of premetastatic niches, the epithelial-to-mesenchymal transition (EMT) and the metastatic cascade through angiogenesis, immunosuppression, and drug resistance [74]. Likewise, FT sEVs may play a role in the progression and spread of FT lesions in the adjacent healthy FT tissue, driving the onset of epithelial ovarian cancer. As such, the alteration of the molecular content of fallopian tube lesion-derived sEVs can potentially indicate disease states and servity as a new class of cancer biomarkers. To establish the path from the non-transformed progenitor cells to the tumor-expressing sEVs, it is necessary to characterize sEVs produced by non-malignant human FTE, which was accomplished using the microphysiological culture system. To begin to predict the candidates of such protein biomarkers, we compared hFTE sEV proteins with STIC tissue proteome and cancer-associated transcriptome profile of hFTE tissue. Currently, there are no reports of EVs from putative preneoplastic lesions, as the direct characterization of STIC-tissue-derived sEVs is challenging. STIC is very small, while proteomics characterization requires the protein abundance within the sample. Further, STIC needs to be diagnosed based on the morphologic and immunohistochemical features of the tissue after fixation. Therefore, comparing hFTE sEV proteins with STIC tissue proteome is a reasonable approach to predict sEV proteins secreted by STIC tissue. Importantly, we cannot ensure that STIC lesions produce sEVs that contain these proteins, but it is of interest that sEVs from FTE can contain some of these proteins as cargo, based on these findings. FT sEV proteins, RAN, FBLN, UBA1 and ALDH2, were only detected in STIC primary fallopian tube tissue lysate, but not in benign fallopian tube. It is possible that these are produced by normal FTE, but are not abundant intracellularly because they are mostly secreted into sEVs. We have also identified FLNA, TUBB, FLNC, and JUP sEV proteins that are overexpressed in secretory cells, which are the cells of origin of most HGSOCs.

Neutrophils mediate innate immunity through the effector mechanism as the first line of defense against foreign pathogens, and are more abundant in the FT compared to the rest of the reproductive tract [75]. Previous studies using flow cytometry have identified neutrophils as one of the most prevalent leukocytes in healthy FT [75,76]. Smith et al. showed that the neutrophils from FT exhibit a distinct phenotype compared to neutrophils in peripheral blood. FT neutrophils express a higher level of vascular endothelial growth factor (VEGF), γ-interferon, CD64, and human class II histocompatibility antigen DR (HLA-DR) compared to blood neutrophils [75]. Molecular classification of the proteins from EV-detected fallopian tube EV proteins associated with neutrophil biology. Myeloperoxidase (MPO) is primarily secreted by neutrophils and is required for neutrophil extracellular trap formation [70]. HLA-G is an inhibitor of neutrophil adhesion to the endothelium, and can potentially bind to ILT4 receptors to suppress neutrophil-mediated immunity. [77] HLA-DRA is detected in FT sEVs in our study, which is not expressed by neutrophils from blood unless stimulated by certain cytokines [78,79,80]. Lysozyme (LYZ) and lactoferrin (LF) are secreted in neutrophil granules for antimicrobial or anti-inflammatory functions [69,81]. The study of the FT neutrophil functions is challenging, as neutrophils tend to undergo apoptosis rapidly ex vivo and extravascular neutrophils are not abundant enough for IHC detection [82]. Microfluidic technology may provide an opportunity to study neutrophil functions in the fallopian tube, as it allows for the long-term culture of fallopian tube explants and can be engineered to sustain a neutrophil culture ex vivo.

The presence of several shared transcripts and the protein cargo of human sEVs from FTE were used to identify 61 cancer-associated proteins in FT sEVs. These 61 FT sEV proteins play an important role in biological processes related to cancer biology and immune response, including RAGE receptor binding, GPER1 signaling, and neutrophil degranulation. RAGE is a multiligand receptor that can propagate cellular dysfunction in a number of pathophysiologic conditions, including cancer [83]. It is expressed in low levels in normal tissue, but its expression increases as the ligand accumulates, which is observed in tumors [84]. The sEV proteins involved in RAGE receptor binding are S100A7, S100A9, and S100A8. S100 protein families are calcium- and zinc-binding proteins that modulate inflammatory processes and immune response [85]. Likewise, Reactome pathway analysis has identified important cellular pathways related to the immune response, including GPER-1 signaling pathway associated with 61 FT sEV proteins. It has been shown that GPER-1 acts as a potential tumor suppressor in ovarian cancer, and its high expression is associated with favorable clinical outcomes [86,87]. The sEV proteins involved in GPER-1 signaling are GNG12 (guanine nucleotide-binding protein G(I)/G(S)/G(O) subunit gamma-12), ITGB1 (integrin beta-1), FN1 (fibronectin), and GNAS (guanine nucleotide-binding protein G(s)). Fibronectin proteins have been reported to facilitate tumorigenesis, angiogenesis, cell migration, and cell adhesion. Lengyel et al. have shown that fibronectin promotes ovarian cancer invasion and metastasis through an α5β1-integrin/c-Met/FAK/Src-dependent signaling pathway, transducing signals through c-Met in an HGF/SF-independent manner [88]. Likewise, Drapkin and the group have shown that L1CAM contributes to the ability of transformed FT secretory cells to detach from the tube and contribute to ovarian cancer pathogenesis by upregulating fibronectin and integrin in malignant cells, and activating AKT and ERK pathways [89]. The upregulation of fibronectin promotes the survival of cell aggregates under anchorage-independent conditions that lose contact with their primary substratum. As such, the alteration of fibronectin protein level in hFTE-tissue-derived sEVs can potentially indicate fallopian tube tumorigenesis.

While these transcript probe sets are selected for their roles in cancer, normal cells also express these transcripts, which was the case for those cultured in this study. Out of 61 cancer-associated FT sEV proteins, we identified several proteins that have a reported role in ovarian cancer or cancer in general, including fibronectin, S100 protein families (S100A7, S100A8, S100A9), integrin (ITGB1), heat shock proteins (HSP90), and versican core protein (VCAN), among others. Future proteomics analysis of FT sEVs in response to known risk factors for high-grade serous cancer tumorigenesis may reveal the important protein cargos that regulate this process. The quantitative analysis of hFTE-derived sEV proteins in normal vs. disease conditions can help predict the state of ovarian cancer carcinogenesis in the FT tube tissue.

Our study identified 295 shared FT sEV proteins from seven patients, which is a relatively small sample size. We did observe patient variabilities in FT sEV protein profile and expression levels among seven biological replicates. Patient 1 had a distinct sEV protein profile, probably due to the unique diagnosis of adenosarcoma. Thus, analysis of FT sEV protein profiles across a larger cohort would be beneficial to determine the representations of sEV proteins in distinct populations and underlying health conditions. Oviduct shifts the EV protein cargos across the estrus cycle under hormonal regulation [90]. However, the fallopian tube cultured in the perfusion system was not exposed to ovarian secretions, such as steroid hormones or growth factors, as in in vivo conditions. Therefore, our model can only represent the FT physiology in menopausal conditions. This limitation can be overcome by exposing FT to ovarian secretions during ovulation, using microfluidic technology, which has been optimized in our lab [27].

## Figures and Tables

**Figure 1 bioengineering-10-00423-f001:**
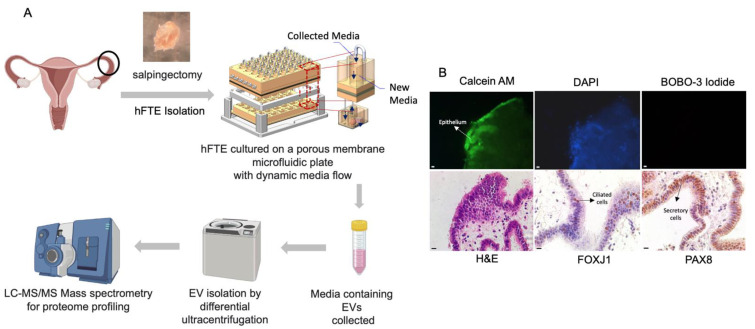
The fallopian tube explants were cultured in a microfluidic platform for sEV collection. (**A**) The workflow of hFTE-derived EV proteomics characterization. (**B**) The live/dead staining of hFTE epithelium cultured in the PREDICT-MOS for six days. The green fluorescence (calcein) indicates live cells, and the red fluorescence indicates dead cells, labeled by BOBO-3 iodide. Immunohistochemistry staining of FOXJ1 and PAX8 confirmed that the hFTE maintained the two major epithelial cell populations, ciliated and secretory cells, after six-day dynamic culture on the PREDICT-MOS. Scale bar = 10 µm.

**Figure 2 bioengineering-10-00423-f002:**
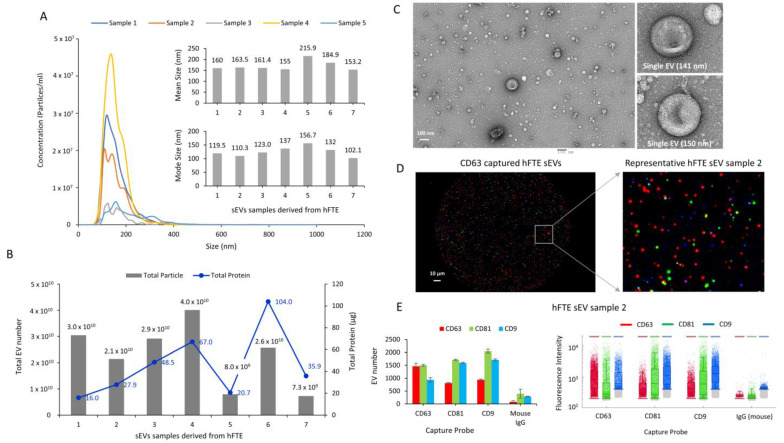
The physical and molecular characterization of sEVs derived from hFTE cultured in the PREDICT-MOS. (**A**) The size profile of sEVs was quantified using nanoparticle-tracking analysis (NTA). NTA graphs of five representative samples are shown. Inset shows the mean size and mode size of sEVs from seven patient samples. (**B**) The total sEV number and total protein content in sEVs quantified via NTA and the Bradford assay, respectively. (**C**) The transmission electron microscopy (TEM) image of sEVs shows cup-shaped morphological properties. Inset shows the micrograph of a single sEVs (Scale bar: 100 nm). (**D**) The image of sEVs captured by the CD63 capture probe in ExoView chip, and detected by CD63 (red), CD81 (green), and CD9 (blue) antibodies (Scale bar: 10 µm). (**E**) The molecular characterization of sEVs quantifying CD63/CD81/CD9-positive sEVs using the ExoView platform. The graph shows the particle number and fluorescent intensity of sEVs captured by the CD63/CD81/CD9 probe, and detected by CD63/CD81/CD9 fluorescent antibodies in representative sEV samples.

**Figure 3 bioengineering-10-00423-f003:**
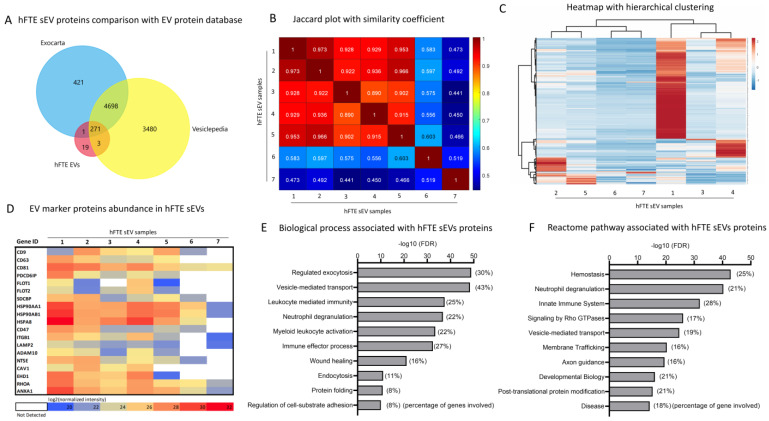
The proteomics characterization of sEVs derived from hFTE cultured in the PREDICT −MOS. (**A**) The comparison of 295 common proteins identified in at least five of seven hFTE sEV samples, with established EV protein databases, Vesiclepedia and Exocarta, showing 275 common proteins and 19 unique proteins. (**B**) The plot referring to the Jaccard index (or similarity coefficient), assessing the similarity in terms of the number of expressed proteins among seven samples based on 295 proteins. A Jaccard index of 1 (diagonal) indicates that the same proteins are expressed (100% similarity). A value of 0 for the Jaccard index would indicate no similarity corresponding to completely different sets of expressed proteins (no overlap or 0% similarity). (**C**) A heatmap demonstrating the protein abundance of 295 proteins among seven hFTE sEV samples. Rows are centered, and unit-variance scaling is applied to rows. Both rows and columns are clustered by Pearson’s correlation as a method, and Ward as a distance. (**D**) A heatmap showing the expression of proteins most frequently identified in EVs, based upon Vesiclepedia and Exocarta databases among seven hFTE sEVs samples. (**E**,**F**) Gene-enrichment analysis using GO biological processes and REACTOME pathway analysis, showing the biological processes and pathways associated with 295 hFTE sEV proteins. Inset in the figure represents the percentage of the genes responsible for a particular biological process.

**Figure 4 bioengineering-10-00423-f004:**
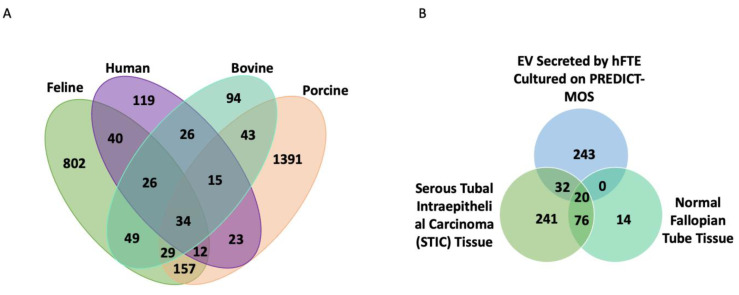
hFTE sEV protein content comparison across species and FT lesions. (**A**) Venn diagram showing the number of sEV proteins identified exclusively in human fallopian tube epithelium (hFTE), bovine, porcine, cat oviduct, or in common [3,33,34]. (**B**) Venn diagram showing shared proteins identified in STIC tissue [35] and proteins found in hFTE-derived sEVs cultured in the microfluidic device, PREDICT-MOS.

**Figure 5 bioengineering-10-00423-f005:**
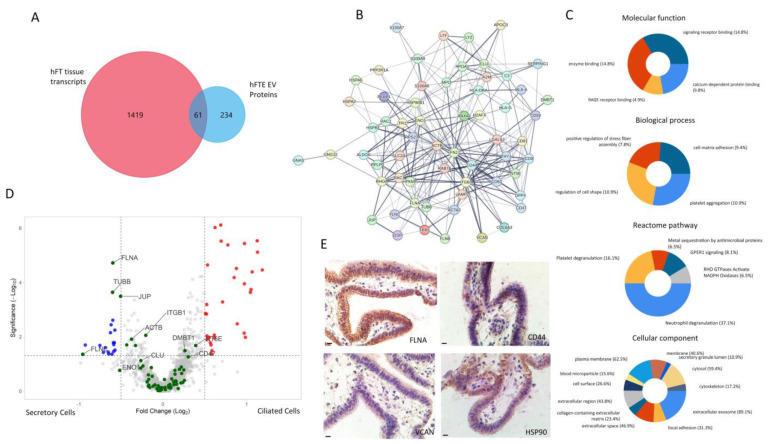
Correlation of fallopian tube tissue transcriptomics profile with a hFTE sEV proteomics profile. (**A**) Comparison between hFTE sEV proteins with hFT tissue transcripts (CTA database, GeoMx) of ciliated and secretory cells identified by GeoMx digital spatial imaging showing 61 common gene IDs. (**B**) Protein−protein interaction network of the 61 common gene IDs via STRING. (**C**) Gene enrichment analysis of 61 common genes showing molecular functions, biological processes, Reactome pathways, and cellular components associated with 61 common genes, using the FUNRICH software. The percentage in the pie chart description represents the percentage of genes out of 61 common genes responsible for particular events. (**D**) The differential expression of genes in ciliated vs. secretory cells in fallopian tube tissue. The green dots represent 61 common gene IDs between the hFTE transcriptome and hFTE sEV proteomics. Four (FLNA, TUBB, JUP, and FLNC) out of 61 genes were significantly upregulated in secretory cells. Blue and red dots are differentially expressed genes in secretory and ciliated cells, respectively. (**E**) IHC staining of representative proteins of 61 common genes expressed in fallopian tube tissue and sEVs. Scale bar = 10 µm.

**Table 1 bioengineering-10-00423-t001:** The list of proteins among the 295 identified human fallopian tube epithelial (hFTE) sEV proteins with known functions in reproductive biology.

Protein Name	Gene ID	Reproductive Function	Source
Plasma membrane calcium-transporting ATPase	PMCA4	Sperm hyperactivation and acrosome reaction	Al-Dossary, A.A. (2013) [25]
Pregnancy zone protein	PZP	Suppress T-cell function to prevent fetal rejection; Efficiently inhibits the aggregation of misfolded proteins during pregnancy	Skornicka, E. (2004) [40] Cater, H.J. (2019) [41]
Profilin 1	PFN1	Actin assembly regulator, essential for cell division and survival during embryogenesis	Witke, W. (2001) [42]
Disintegrin and metalloproteinase domain-containing protein	ADAM10	Regulate Notch signaling, early embryonic cardiovascular development	Zhang, C. (2010) [43]
Gelsolin	GSN	Actin-severing protein regulates acrosome reaction and sperm capacitation	Finkelstein, M. (2010) [44]
Tetraspanin CD9	CD9	Egg membrane protein essential for sperm-egg fusion	Miyado, K. (2000) [45]
Heat shock 70 kDa protein 2	HSPA2	Sperm fertilization ability	Feng, H.L (2001) [46]
Heat shock 90 kDa protein	HSP90	Modulate sperm capacitation via Erk1/2 and p38 MAPK signaling	Sun, P. (2021) [47]
Tubulin beta-4B chain	TUBB4B	Ciliary motility; egg, gamete transport	Yuan, S. (2021) [48]
Serpin Family G Member 1	SERPING1	Endometrial receptivity, implantation	Mirkin, S. (2005) [49]
Annexin A2	ANXA2	Critical for embryo adhesiveness to the human endometrium by RhoA activation	Garrido-Gómez, T. (2012) [50]

**Table 2 bioengineering-10-00423-t002:** The list of proteins among the 295 identified human fallopian tube epithelial (hFTE) sEV proteins with known functions in cancer progression and development.

Protein Name	Gene ID	Function	Source
CD44 antigen	CD44	Induce cell proliferation, increase cell survival and cellular motility	Senbanjo, L. (2017) [51]
Ras-related nuclear protein	RAN	Promote membrane targeting and stabilization of RhoA to orchestrate ovarian cancer cell invasion	Zaoui, K. (2019) [52]
Integrin beta	ITGB1	Promote ovarian tumor progression and metastasis	Akinjiyan, F. (2022) [53]
S100 calcium binding protein A6	S100A6	Predict peritoneal tumor burden and is associated with advanced stage in ovarian cancer	Wei B. (2009) [54]
Versican core protein	VCAN	Modulate cell adhesion, proliferation, apoptosis, angiogenesis, invasion, and metastasis	Russo, A. (2022) [27]
Periostin	POSTN	Ovarian cancer migration and adhesion, wound healing	Yue, H (2021) [55]
The Ras homologous (Rho) protein	RHOs	Promote ovarian cancer progression and chemoresistance	Jeong, K.J. (2012) [56] Sharma, S. (2014) [57]
Transforming growth factor, beta-induced	TGFBI	Promote ovarian cancer migration and contributes to an Immunosuppressive Microenvironment	Lecker S.M.L. (2021) [58]
Aldehyde dehydrogenase 1A1	ALDH1A1	ALDH1A1 maintains ovarian cancer stem cell-like properties by altered regulation of cell cycle checkpoint and DNA repair network signaling	Meng, E. (2014) [59]
Lysosome-associated membrane glycoprotein 2	LAMP2	Regulate lysosomal stability as well as in autophagy	Eskelinen, E. (2002) [60]
Ras-related protein Rap-1b	RAP1B	The member of RAS oncogene family, promotes ovarian cancer metastasis via notch signaling	Lu, L (2016) [61]
Ras-related protein Rap-1A	RAP1A		
Fibronectin	FN1	Promote ovarian cancer invasion and metastasis through an α5β1-integrin/c-Met/FAK/Src-dependent signaling pathway, transducing signals through c-Met in an HGF/SF-independent manner	Lengyel, E (2010) [62]

## Data Availability

The datasets analyzed during the current study are available from the corresponding author on reasonable request.

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
