# Peer review of "Proteomic Profiling of Fallopian Tube-Derived Extracellular Vesicles Using a Microfluidic Tissue-on-Chip System"

_bioengineering, 2023, doi:10.3390/bioengineering10040423_

Round 1

Reviewer 1 Report

Didi Zha and colleagues evaluated the proteins present in cultured fallopian tube cells. In addition, spatial transcriptomics was performed on the fallopian tubes. This is the first time fallopian tube exosomes have been characterized. The manuscript would benefit from som major changes:

1. Validation of some of proteomics findings using different method

2. description of how done and results from transcriptomics

3. Results section of manuscript needs to be revised to only contain results and not discussion. 
4. the last figures are comparing results with previously published work and alot of speculation and discussion instead of results clear and concise

5. patient demographics

The authors should also consider providing access to all this data and analysis for all to use. This would be used by so many studying fertility, fibroids, endometriosis, LGSOC, HGSOC, clear cell carcinoma, mucinous ovca, carcinosarcoma etc. 

Author Response

Didi Zha and colleagues evaluated the proteins present in cultured fallopian tube cells. In addition, spatial transcriptomics was performed on the fallopian tubes. This is the first time fallopian tube exosomes have been characterized. The manuscript would benefit from som major changes:

  1. Validation of some of proteomics findings using different method

Response: We thank reviewer 1 for the comments of using orthogonal methods to validate proteomics findings of fallopian tube-derived extracellular vesicles.

We have validated some EV proteins from fallopian tube using the following methods:

  1. We have validated presence of tetraspanins (Cd63, CD9, and CD81) using Exoview R100 platform as shown in the figure above. Exoview is highly sensitive automated platform that provide high resolution sizing, counting, and biomarker detection at the single EV level. The working principle of Exoview is similar to sandwich ELISA shown in figure above. Exoview chips contains three different capture antibodies (CD9, CD63, CD81) which binds to respective tetraspanins in EVs. Immobilized EVs are detected by fluorescent antibodies using microscopy. The presence of red, green, and blue puncta in Figure 2D of manuscript validates the presences of CD63, CD81, and CD9 respectively. Use of Exoview platform is reported by several literature for highly sensitive validation of EV proteins with single-EV sensitivity (1-3) referenced below. Using this platform, we have validated presence of EV marker proteins in 4 EVs samples (biological replicates) and reported data from one representative samples in the manuscript (Figure 2D&E).

  1. The comparison with known EV protein databases.

We have compared our proteomics data with two extracellular vesicle databases, Exocarta and Vesiclepedia, which contain EV proteins identified in human cell lines and body fluids-derived extracellular vesicles. The considerable overlap (93%) of our EV proteomics findings vs. two known EV molecular databases can serve as a validation of our EV isolation and detection techniques.

  1. The detection of EV proteins present in primary human fallopian tube using immunohistochemistry.

To confirm the origin of EV proteins we detected, we have validated four (CD44, FLNA, Versican and HSP90) of the 61 EV proteins encoded by fallopian tube transcripts in tissue, from which the EVs were isolated, using immunohistochemistry.

We would also like to emphasize that the scope of the study is for discovery proteomics to establish a baseline proteomic profile of FT tissue explant. While we have validated some EV proteins using different method, the comprehensive proteomic profile identified by mass spectrometry can serve as valuable resource for scholars in the field.

References:

  • Breitwieser, Kai, Leon F. Koch, Tobias Tertel, Eva Proestler, Luisa D. Burgers, Christoph Lipps, James Adjaye, Robert Fürst, Bernd Giebel, and Meike J. Saul. "Detailed Characterization of Small Extracellular Vesicles from Different Cell Types Based on Tetraspanin Composition by ExoView R100 Platform." International Journal of Molecular Sciences 23, no. 15 (2022): 8544.

  • Mahida, Rahul Y., Joshua Price, Sebastian T. Lugg, Hui Li, Dhruv Parekh, Aaron Scott, Paul Harrison, Michael A. Matthay, and David R. Thickett. "Extracellular Vesicles in Lung Health, Disease, and Therapy: CD14-positive extracellular vesicles in bronchoalveolar lavage fluid as a new biomarker of acute respiratory distress syndrome." American Journal of Physiology-Lung Cellular and Molecular Physiology 322, no. 4 (2022): L617.

  • Donzelli, Julia, Eva Proestler, Anna Riedel, Sheila Nevermann, Brigitte Hertel, Andreas Guenther, Stefan Gattenlöhner, Rajkumar Savai, Karin Larsson, and Meike J. Saul. "Small extracellular vesiclederived miR5745p regulates PGE2biosynthesis via TLR7/8 in lung cancer." Journal of Extracellular Vesicles 10, no. 12 (2021): 12143.

  1. description of how done and results from transcriptomics

Response: We thank reviewer 1 for the suggestions of strengthening the descriptions of transcriptomic analysis.

For spatial transcriptomics analysis, we first used a library of RNA probes (~1800 genes) with a photocleavable linker (oligo tags) to perform in situ hybridization on fallopian tube tissue slides. Next, we used fluorescent antibodies to label ciliated cells and secretory cells. Based on the fluorescent labels, we selected region of interests containing ciliated or secretory cells for analysis. This allows us to collect the transcriptomics data from these two cell populations separately. The slides were exposed to UV light, and the oligo tags were collected for next-generation sequencing. The reads of these oligo tags on next-generation sequencer reflect the abundance of their corresponding RNA probes and original RNA content present in the fallopian tube tissue. We have identified 1480 transcripts in hFTE tissue explants from GeoMx® Cancer Transcriptome Atlas containing a library of 1800 genes, among which, 61 genes encode for proteins which are sorted and enriched in fallopian tube-derived EVs. We also identified four (FLNA, JUP, FLNC and TUBB) fallopian tube transcripts encoding EV proteins that are upregulated in secretory cells, which indicate they encode secretory cell-enriched fallopian tube EV proteins.

I have added the description in method section 2.3 in the original text highlighted in yellow.

  1. Results section of manuscript needs to be revised to only contain results and not discussion. 

Response: We have moved the discussion on the results presented in figure 5 to the discussion section (highlighted in yellow in the third last paragraph of discussion).

  1. the last figures are comparing results with previously published work and alot of speculation and discussion instead of results clear and concise

Response: We thank reviewer 1 for the suggestion on revising the result section of Figure 4. The following is a brief summary of the results of panel A and B of Figure 4, and we have highlighted the change of this section in yellow.

Panel A:

To determine the differences between animal models used to study oviductal EV proteins, we compared human fallopian tube-derived EV protein profile with those previously reported in mammals. Based on this comparison, bovine oviduct model represents the most human-relevant extracellular vesicle physiology to fallopian tube. In addition, human has a unique fallopian tube EV biology compared to other mammals, indicated by 119 EV proteins exclusively detected in human.

We have moved the following sentences to the second paragraph of the discussion section (highlighted in yellow) to make the results in Figure 4 clearer and more concise.

“119 proteins were exclusively detected in human fallopian tube-derived sEVs but no other mammalian species analyzed…”

Panel B

To predict EV proteins indicative of lesions in fallopian tube, we compared benign fallopian tube-derived EV proteins with proteins present in the diseased fallopian tube (serous tubal intraepithelial carcinoma, STIC) tissue. We have identified 32 EV protein candidates that are only expressed by primary fallopian tube tissue with STIC lesion but not benign fallopian tubes.

We believe these comparisons are crucial to the findings of this study, because they revealed 1) the unique human fallopian tube biology compared mammals, 2) bovine oviduct as the most human-relevant model to study oviduct EV biology, and 3) potential fallopian tube EV proteins indicative of early lesion of ovarian cancer in the fallopian tube.

Characterizing EV proteins secreted by fallopian tube STIC lesions alone is challenging, because STIC is very small, and proteomics relies on abundance within a sample and has no amplification steps that are useful in other methods like RNA-seq. Further, there would be no way to collect EV from that small lesion in vivo and patients are rarely diagnosed with ovarian cancer when they only present with a STIC lesion, instead it is usually found along with peritoneal disease. The paper we included is the only report on proteins expressed in STIC lesions. Therefore, comparing EV proteins secreted by benign fallopian tube with proteins present in primary STIC tissue is a reasonable method to predict the protein cargos of EVs secreted by fallopian tube STIC lesions.

We have revised the included this discussion in the third paragraph of the discussion section highlighted in yellow.

“Currently, there are no reports of EVs from putative preneoplastic lesions, as the direct characterization of STIC tissue-derived sEVs is challenging…”

  1. patient demographics

The authors should also consider providing access to all this data and analysis for all to use. This would be used by so many studying fertility, fibroids, endometriosis, LGSOC, HGSOC, clear cell carcinoma, mucinous ovca, carcinosarcoma etc. 

Response: We thank reviewer 1 for the comment on the data availability.

We have uploaded the raw data for proteomics and transcriptomic study in the journal.

We have included patient demographics (age, diagnosis) in the Supplementary Table 1.

Regarding English language, we have extensively reviewed the manuscript by multiple colleagues including native English-speaking colleague. We have also use Grammarly software to remove any grammatical errors.

Reviewer 2 Report

The manuscript is very interesting and has a novelty of FF, especially the importance of successful fertilization.  It is suitable for publication as it is. 

Author Response

The manuscript is very interesting and has a novelty of FF, especially the importance of successful fertilization.  It is suitable for publication as it is. 

Response: We thank reviewer 2 for the comment and we also believe these findings can provide insights into sEV-mediated mechanisms in fallopian tube reproductive biology for the future studies.

Reviewer 3 Report

The authors of the manuscript titled-" Proteomic Profiling of Fallopian Tube-derived Extracellular 2 Vesicles Using a Microfluidic Tissue-on-Chip System" have presented one of the unique studies in the area of Ovarian Cancer biology where the oviductal secretions play a significant role. Even it is known about their role, but the molecular insights are less known. This study represents a comprehensive approach of isolation of oviductal proteins using a microfluidic based system in which the authors are working and also have established a system. I am happy to see the design, results and discussion of their data. The study sounds good and is having a huge application in coming time. The oviductal secretions from the epithelial cell is not only complex in composition but also they display very complex interplay crosstalks with the cilliary epithelium as well as play significant role during fertlisation. Therefore, a crtical and exhaustive analysis of these EVS will provide novel insights into the actions of these EVs. I recommend the consideration of this manuscript with the consideration of some of my suggestions. 

1. Abstract: Well written.

2. Introduction may be reduced as it is containing some of the generalized information and at this stage they require no attention.

3. Methodology is clear but the major concern is small sample size which need to be more. I think, for this type of study two or more groups may be considered to compare the EVSs and their proteomic profile. I suggest the authors to consider the oviductal EVS of at least of two origins like diseased and non diseased for better comparison and sloid conclusions. 

4. Results: Nicely written.

5. Discussion appears to be fine. Little more literature exploration is required.

overall the findings are good and novel.

Author Response

The authors of the manuscript titled-" Proteomic Profiling of Fallopian Tube-derived Extracellular 2 Vesicles Using a Microfluidic Tissue-on-Chip System" have presented one of the unique studies in the area of Ovarian Cancer biology where the oviductal secretions play a significant role. Even it is known about their role, but the molecular insights are less known. This study represents a comprehensive approach of isolation of oviductal proteins using a microfluidic based system in which the authors are working and also have established a system. I am happy to see the design, results and discussion of their data. The study sounds good and is having a huge application in coming time. The oviductal secretions from the epithelial cell is not only complex in composition but also they display very complex interplay crosstalks with the cilliary epithelium as well as play significant role during fertlisation. Therefore, a crtical and exhaustive analysis of these EVS will provide novel insights into the actions of these EVs. I recommend the consideration of this manuscript with the consideration of some of my suggestions. 

  1. Abstract: Well written.
  2. Introduction may be reduced as it is containing some of the generalized information and at this stage they require no attention.

Response: We thank reviewer 3’s comments on the revision of introduction.

Response:We have reduced the background content regarding tumor cell-derived EVs in the introduction section.

  1. Methodology is clear but the major concern is small sample size which need to be more. I think, for this type of study two or more groups may be considered to compare the EVSs and their proteomic profile. I suggest the authors to consider the oviductal EVS of at least of two origins like diseased and non diseased for better comparison and sloid conclusions. 

Response:We thank reviewer 3’s comment on the study design.

Response:Although the fallopian tubes we used in the study are pathologically benign, the patients donating their fallopian tube are doing so because they have been recommended for surgery based on other gynecological conditions. It is challenging to obtain fallopian tube from a healthy individual without any underlying conditions. We understood that these conditions may cause certain discrepancy compared to fallopian tube EV physiology of a healthy individual, which we have address is the discussion section as the limitation of this study.

  1. Results: Nicely written.
  2. Discussion appears to be fine. Little more literature exploration is required.

Response: We have revised the discussion with more literature insights in line highlighted in yellow in the discussion section.

overall the findings are good and novel.

Round 2

Reviewer 1 Report

The manuscript by Didi Zha et al describes the proteomics profiling of fallopian tube EVs fills a gap in the literature and an identifies proteins that can further explored in fertility, cancer etc. The results are clear and concise.